# Enhancing the Thermal Images of the Upper Scarp of the Poggio Baldi Landslide (Italy) by Physical Modeling and Image Analysis

**Andrea Massi** [1], **Michele Ortolani** [2,*], **Domenico Vitulano** [3], **Vittoria Bruni** [3] and **Paolo Mazzanti** [4]

1    Dipartimento di Fisica, Sapienza University of Rome and NHAZCA srl, 00185 Rome, Italy
2    Dipartimento di Fisica, Sapienza University of Rome, 00185 Rome, Italy
3    Dipartimento di Scienze di Base e Applicate per l'Ingegneria, Sapienza University of Rome, 00185 Rome, Italy
4    Dipartimento di Scienze della Terra e Centro di Ricerca CERi, Sapienza University of Rome, 00185 Rome, Italy
*    Correspondence: michele.ortolani@uniroma1.it

**Abstract:** We present new methods for physical interpretation and mathematical treatment of the imaging contrast observed in thermal images of the rocky upper scarp of the Poggio Baldi landslide (Italy), which is part of a natural laboratory. Exemplar thermal images have been acquired with a high-performance camera at a distance of around 500 m, in a geometry where reflection is expected to dominate over thermal emission. The digital pixel intensities have therefore been considered as wavelength-integrated infrared spectral reflectance, irrespective of the temperature scale loaded into the camera software. Sub-portions of the scarp producing a lower signal have been identified by a multiscale image segmentation algorithm and overlaid on the visible image to provide an interpretation for the different thermal imaging contrast mechanisms that may be exploited for landslide monitoring in the future.

**Keywords:** landslide monitoring; thermal imaging; image segmentation; blackbody radiation law

## 1. Introduction

The fast remote detection of landslides and rockfalls impacting on roads, railway tracks, electric grids and other infrastructures has become a crucial safety and efficiency issue [1]. Hydro geological models [2], airborne [3] or satellite imaging [4] and the related image processing techniques are typically employed to construct landslide hazard maps, but these are of little help when it comes to detecting actual events: there is an obvious need for ground-based real-time monitoring of critical infrastructures that can provide alarm signals for the population and infrastructure safety. In the case of rockfalls hitting transport infrastructure, the alarm time should be a fraction of a minute, but it could be also possible to detect in advance the triggering events of a landslide/rockfall, such as anomalous water content after heavy rainfall and/or small-scale movements of the ground preceding a more extensive ground movement [5]. In many cases, the early detection of triggering events may be sufficient for executing prompt intervention actions. Therefore, there is a clear need for real-time monitoring techniques applied to rocky scarps, slopes, cliffs, scarps that are part of landslides [6].

Parameters such as water content and micro-movements of the ground can be detected with pervasive sensor networks [7], including mechanical stress sensors, humidity sensors, accelerometers, and atmospheric pressure sensors. These sensor networks may be low-cost and long-lived, but their installation and maintenance still require costly, and sometimes risky, personnel working time. Therefore, remote sensing and imaging techniques based on cameras and radars have become popular in recent years [8,9]. Automatic Change Detection (CD) analyses of images collected by cameras is one of the most promising solutions for this

purpose [10–12]. At present, the most common components for this application are visible-light/near-infrared (VIS/NIR) cameras and microwave radars. The main disadvantage of VIS/NIR cameras over radars is that they are almost blind in the darkness and through fog. On the other hand, radars are based on active microwave emitters and detectors that can "see" through fog, darkness and bad weather conditions but they cannot provide a view of natural landscapes [13]. Interferometric "imaging" radars (for example synthetic aperture radars, SAR) can detect and localize changes in the position of corner reflectors and/or of specific points of a rock slope with a broad field of view ([10–14]); however, they cannot provide clear and understandable images of natural landscapes, thus complicating the geological interpretation of the monitoring data.

In between VIS/NIR light and the microwaves sits the electromagnetic range of thermal infrared (TIR), with the especially important remote-sensing atmospheric transmission window of wavelengths $\lambda$ between 8 and 13 μm. TIR radiation is naturally emitted by the environment at room temperature at any time of the day/night and without the need for sunlight or active laser/radar illumination. The lack of an illumination source eliminates the problem of shadows changing with time that plagues automatic CD with VIS/NIR cameras [15]. In fact, TIR vision systems are already employed at crossing points of roads and railway tracks to detect persons, animals and vehicles (which are all sources of heat, and therefore of TIR radiation well above the environmental TIR emission); however, here we discuss the completely different (and more complicated) problem of geological hazard monitoring with TIR imaging. If compared to heat-source detection, TIR images of natural landscapes display weaker contrast driven by different factors: temperature inhomogeneity, surface roughness, physical-chemical composition influencing the optical properties. Indeed, the TIR image contrasts are totally different from those observed in the VIS/NIR [16–20]. Finally, the penetration of TIR radiation through fog and rain is typically better than that of VIS/NIR light [9], although certainly not comparable to that of radars [21].

TIR imaging has been previously applied to rocky scarps with the aim of identifying structural features that may be hidden to VIS/NIR radiation [22]. In Ref. [17], rocky bridges hidden by large granite flakes have been located by measuring the TIR emission profile across the flakes, which is modified by the thermal conductance of the hidden rocky bridge. In Refs. [8,23,24], TIR imaging has been employed to detect the presence of faults, cracks, and eroded zones in natural and artificial scarps adjacent to roads. An automated algorithm for TIR image analysis of weakness points in a rock cliff has been presented in Ref. [25].

In this article, we discuss the feasibility of a specific landslide monitoring application taking the Poggio Baldi landslide (Italy) as a research example. The landslide and its geological map are shown in Figure 1 and Figure 2. Using the high-performance TIR camera in Figure 1b (which is still less expensive than an imaging radar), a monitoring distance of around 500 m was demonstrated. The high-performance TIR camera objective has a very broad field of view of the upper scarp, comparable to that of a standard VIS/NIR imaging camera, which enables the overlap of real-time analyzed TIR images and archived VIS/NIR images. We experimentally showed that the relevant TIR imaging contrast can be evidenced by rigorous image analysis based on multiscale image segmentation algorithms, which have seldom been applied to landslide monitoring, and only to standard VIS/NIR imaging [1].

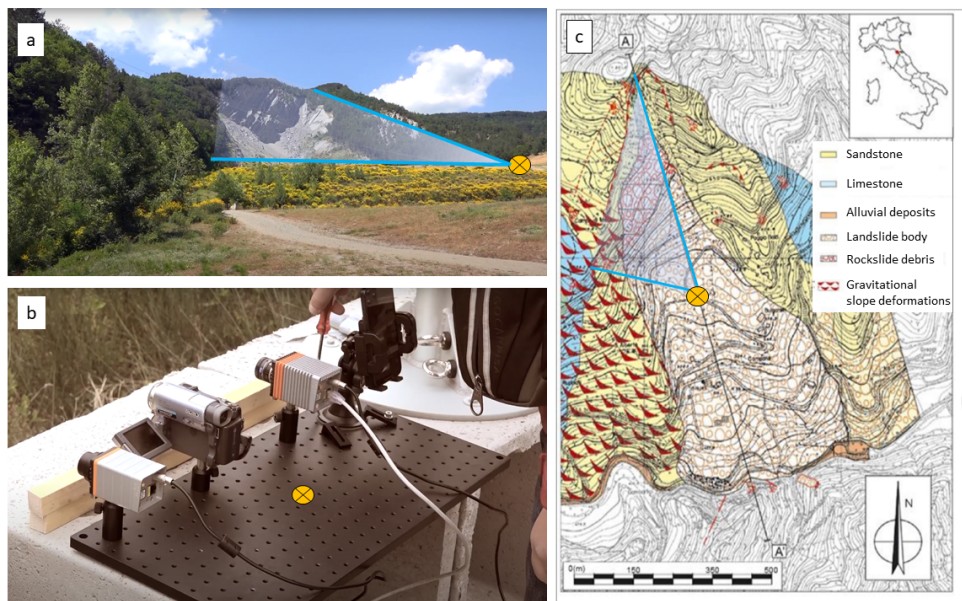

**Figure 1.** (**a**) Photo of the landslide with its surrounding environment. (**b**) Optical system used for multi-wavelength imaging. From left to right: Thermal infrared (TIR) camera, low- resolution visible camera for image selection and alignment among sensors, short wavelength IR (SWIR) camera, and smartphone for high-resolution visible imaging. (**c**) Geological map of the Poggio Baldi landslide area; [26] modified from [27]. In each panel, a yellow black-crossed dot indicates the position of the optical system including the TIR camera.

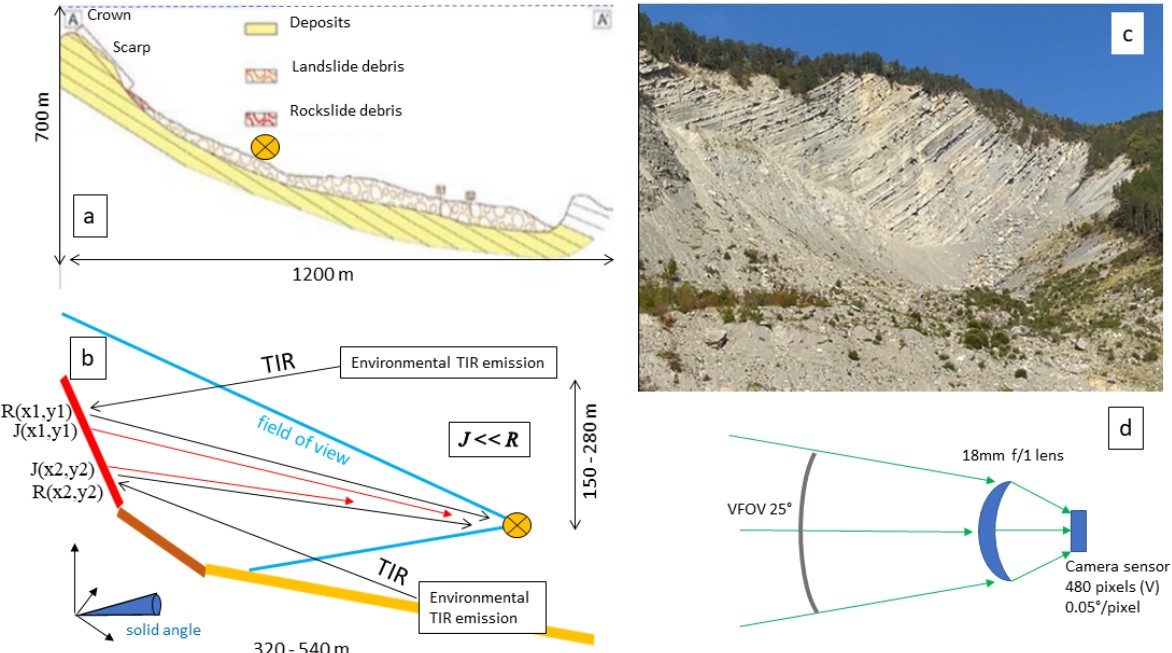

**Figure 2.** (**a**) Cut view of the landslide and its constituents with the distance of the zone that is affected by the landslide and the height of the landslide. (**b**) Theoretical model of the reflection and the emission of the elements of the landslide projected in the pixel (x,y) of the TIR camera detector with the distance and height of the detector respect the landslide. (**c**) Area corresponding to the field of view of the TIR camera, cropped from a photograph in the visible range taken with a smartphone in Figure 1c. (**d**) Geometrical optics model of the vertical field of view of the TIR camera.

## 2. Landslide Description

The Poggio Baldi landslide is one of the largest rock and debris phenomena in the Italian Apennines and a permanent natural monitoring site managed by the Department of Earth Sciences of the Sapienza University of Rome [27]. Advanced remote sensing tools are installed in order to monitor the activity of the main rocky scarp, which is often a source of rockfalls, and the related debris talus at the toe [26]. The first activation dates to 25 March 1914, whereas the last reactivation of the landslide deposit took place on 18 March 2010, and it was triggered due to the increase of water pressure in the pores of the paleo-landslide's body following the melting of the snow blanket caused by a sudden increase in temperatures. The landslide has an estimated volume of about $4 \times 10^6$ m$^3$, and it is currently active in its upper scarp due to frequent rockfall events.

The slope is a marly sandstone formation from the Miocene, involving an alternation of claystone, siltstone and sandstone strata, arranged in a monoclinal dip slope sequence. A slight bending of the strata occurs in the lower part of the slope: the bedding attitude, dipping at about 45$^o$ slope, progressively decreases reaching dip angles of about 15–20 downstream. The main rock slope, which is the target of this study, is a sub-vertical scarp with a rise of about 100 m and a width of 250 m, and it is characterized by high-frequency rockfall processes. Since the first activation in 1914, despite the following steady state of the main landslide body, the newly formed vertical rock cliff has always been frequently affected by rockfalls. The instability is predisposed by geologic, geomorphological and structural factors, such as the alternation of arenaceous and clayey strata and the presence of several discontinuities. Moreover, considering the progressive erosion of the clay strata, which gradually increases the overhang of the arenaceous strata, rockfalls are mainly triggered by: (i) the breaking of the arenaceous mass due to shear strength surplus and (ii) the intersection of a joint with the exposed surface of the cliff. The detached debris and blocks often find deposition surfaces over the underlying arenaceous strata. Over time, the volume of fallen material increases on the overhanging strata until the debris friction angle (39 degrees) is exceeded, with the consequent remobilisation. These conditioning factors, combined with each other, are the main cause of the widespread instability throughout the vertical rock cliff. As a matter of fact, the general loss of volume from the vertical rock cliff is estimated to be in the range of 2.0 to $2.8 \times 10^3$ m$^3$ per year.

## 3. Physical Model

It is generally believed that thermal infrared imaging can only identify direct sources of heat, such as living beings or engines in operation, or, to enter the domain of geological risk monitoring, volcanic eruptions [9–28] or geothermal heat sources [29]. In the specific case of landslide monitoring, infrared thermography has been used to characterize either the different types of terrains within a landslide area [13,15] or the thermal and mechanical properties of rocky walls [8,17,30], as we also do here. Up to now, data have been generally interpreted by invoking the Stefan–Boltzmann law, for which the TIR power emitted by an object is proportional to the fourth power of the temperature $T$ through the emissivity $\epsilon$:

$$J(x,y) = \epsilon \cdot \sigma T^4(x,y), \tag{1}$$

where $\sigma$ is the Stefan–Boltzmann constant. The emissivity $\epsilon$ is usually set to 1 for all pixels in the image (blackbody limit). Based on this law, thermal images produced by TIR cameras are usually read with an absolute temperature scale provided by the camera manufacturer, together with some kind of absolute calibration protocol. This "blackbody approximation" is often overused [13]: the temperature of the objects contained in each pixel of the image, alone, does not necessarily determine the TIR radiation power incident on the TIR camera (except, of course, in those cases where the thermal emission totally dominates over the other components of radiation such as in the case of the thermal imaging of volcanic lava [28]). In particular, if the emissivity of water-rich biomass (including living beings, vegetation and wet organic soil) could be described by the blackbody law

($\epsilon = 1$), or by the graybody-contrast approximation (material-dependent but wavelength-independent $\epsilon < 1$), the spectrum of crystals, minerals, and rocks strongly depends on the TIR wavelength $\lambda$ and it is precisely defined by the exact value of optical constants of the specific mineral. For a given object volume and surface orientation, the integrated spectral emittance $e(\lambda)$, displaying a material-specific dependence on the TIR wavelength in the $\lambda \approx 8 - 13\mu m$ range, defines the parameter $\epsilon$ in Equation (1) as follows:

$$\epsilon = \frac{1}{\Delta\lambda} \int_{8\mu m}^{13\mu m} e(\lambda)d\lambda, \tag{2}$$

where $\Delta\lambda$ is the spectral range of integration for TIR remote sensing and $e(\lambda)$ is the adimensional product of the absorption coefficient $\alpha(\lambda)$, also called the IR fingerprint of the material, and a penetration length scale of TIR radiation in the material, which is of the order of 1 μm for rocks and minerals. In this work, we approximate the spectral dependence of Planck's blackbody emission law at T~300 K as almost wavelength-independent in the 8–13 μm range. This considerable conceptual simplification is justified because the Planck's law at $T = 300$ K peaks at around $\lambda = 10$ μm (about 30% of the total TIR emitted by a blackbody at $T = 300$ K lies in this relatively narrow wavelength range), but it should be noted that it would be straightforward to include the Planck's law in our model to increase its quantitative accuracy. In the case of a rocky scarp almost free of biomass, it is possible to obtain, from a TIR image, information on the different materials contained in the different pixels of the image (different rocks, sand, minerals, water, or a combination of these), not from their thermal properties (i.e., from the equilibrium temperature that they assume given their thermal capacitance and conductivity) but from their optical properties (their different TIR emissivity and/or, as we shall see, reflectivity). This identification is especially accurate if one can assume temperature homogeneity across the TIR image:

$$J(x,y) = \epsilon(x,y) \cdot \sigma T^4(x,y) \approx \epsilon(x,y) \cdot \sigma T^4, \tag{3}$$

where the last approximation (homogeneous temperature) almost certainly holds in the case of the present work, because the exposure of the scarp to sunlight is homogeneous and the thermal conductivity along the marly-arenaceous strata is expected to be quite high, thereby forcing a very similar $T$ for all portions of the scarp.

A second important correction to be made to the simple Stefan–Boltzmann emission model comes from the reflected TIR power. A steep scarp made of clean rocks mostly free of vegetation and soil, such as the Poggio Baldi landslide, with constant average orientation angle with respect to the vertical direction over the entire inspection area, will reflect most of the TIR power that hits it towards approximately the same specular direction. TIR power is emitted from neighboring terrain, rock and vegetation in the form of their own graybody radiation (hereafter called "environmental thermal sources") and, in daylight, by the sun. As discussed in detail in Section 4 below, we estimate that, in the present case of a steep rocky scarp and a TIR camera objective with a broad field of view (fov), the reflected TIR power $R(x,y)$ can easily overcome the TIR power emitted by the element of the scarp itself $J(x,y)$. We can therefore write for the total TIR power $P(x,y)$, reaching each pixel of the camera:

$$P(x,y) = J(x,y) + R(x,y) \approx R(x,y), \tag{4}$$

where $R(x,y)$ is defined by another optical property of the materials contained in each element of the scarp corresponding to a pixel in the TIR image: the spectral reflectance $r(\lambda)$. The spectral reflectance is distinct from the spectral emittance $e(\lambda)$ but it can be derived from the same physical model of the optical constants, called the Lorentz oscillator model. In general, $R(x,y)$ can be calculated from the known $r(\lambda, x, y)$ of the materials in each pixel, from the solid angle fraction $\int d\Omega$ corresponding to the number of specular reflection paths connecting the environmental thermal sources to the reflecting surface and then to the TIR camera within its FOV, and from the environmental temperature $T_{env}$, which is obviously the same for all pixels in the image. For an ideal scarp with constant inclination across the

inspection area, we can assume that the integration domain in $d\Omega$ is the same for all pixels corresponding to the scarp, and we can reduce the $x, y$ dependence to $r(\lambda, x, y)$:

$$R(x,y) = \frac{1}{2\pi^2} \frac{1}{\Delta\lambda} \int_{fov} d\Omega \int_{8\mu m}^{13\mu m} r(\lambda, x, y) \cdot \sigma T_{env}^4 d\lambda, \qquad (5)$$

and therefore the TIR image becomes a map of the optical properties of the materials forming the surface of the scarp. In the present case, however, the overhanging arenaceous strata constitute sub-portions of the scarp with completely different orientation angles; therefore, they will reflect less TIR radiation towards the camera if compared to steep sub-portions, as we shall see. For each element of the scarp imaged by one pixel of the camera, one should therefore adjust the solid-angle integration domain $\Omega(x, y)$, quantifying the reduced number of specular reflection paths within the FOV. In general, less reflected TIR power is expected from sub-portions of the scarp with an anomalous inclination angle, such as the overhanging strata of the present scarp.

In Figure 3a, we show the spectral emittance $e(\lambda)$ and reflectance $r(\lambda)$ calculated from the optical constants of argillaceous sandstone, which were obtained with a fit to the Lorentz oscillator model of the diffuse reflectance spectrum reported by the NASA-Ecostress spectral library (sample No. Ward63). One can see two prominent spectral peaks at 8.3 and 9.2 μm wavelength corresponding to two strong vibrational modes of the Si-O-H octahedral network of the silicate rock. A third smaller peak is seen at 12.5 μm, related to vibrations involving the cation (Fe, Al, Mg...). The specific spectral shape of the reflectance of sandstone in Figure 3a was considered in this work as the relevant spectral dependence for the calculation of TIR image contrast.

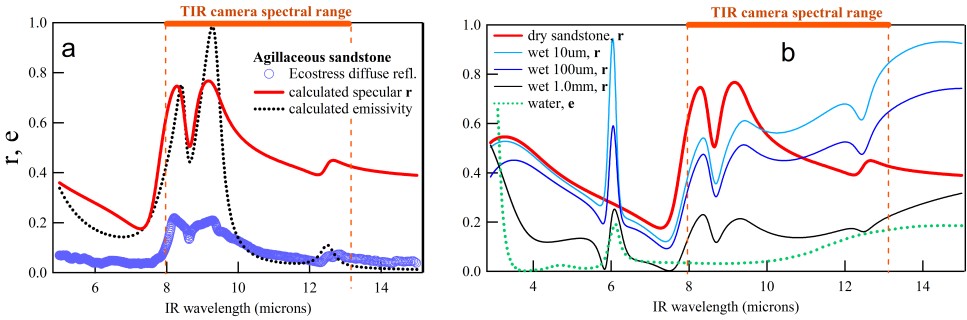

**Figure 3.** a. Blue circles: diffuse reflectance data (spectral signature) of the argillaceous sandstone (sample No. Ward63 from the ECOSTRESS Spectral Library); calculated specular reflectance (red curve) and emittance (black dotted curve) of the same argillaceous sandstone sample. The Lorentz oscillator model has been used to fit the data and to perform the calculations. b: calculated reflectance of dry argillaceous sandstone (red curve, same as left panel), with 10, 100 μm of water on top of the rock surface (light, dark cyan line) and with 1 mm of water on top (black line). The green dotted line is the emittance of liquid water (green dots).

As we shall see below, in our images we have surprisingly observed different TIR signal levels from neighboring portions of the scarp featuring nominally identical mineral compositions. In the case of overhanging arenaceous strata, this contrast can be tracked down to a different spatial orientation impacting on the total reflected TIR power towards the detector. In other cases where the different spatial orientation cannot be invoked, we hypothesize that the lower TIR signal may correspond to a thin layer of liquid water inhomogeneously covering the rocky surface. This hypothesis is also based on the geological history of the Poggio Baldi landslide (2010 event triggered by water pressure increase [26]). In other words, we propose that TIR imaging may be used to identify the scarp locations that are more wet, or more prone to water absorption and retention, hence possibly to (past or future) rockfall [31,32]). To demonstrate this hypothesis, we have produced a multi-layer optical model of the reflectance of an infinitely thick sandstone layer coated with a very

thin water layer, which we show in Figure 3b. It is apparent that the presence of a water layer of 100 μm thickness or more can sensibly decrease the observed spectral reflectance integrated in the 8–13 μm range. This peculiar spectral behavior derives from the complex interplay between the sandstone optical constants and the optical constants of liquid water. It can be understood in terms of a refractive index matching effect: the high reflectance of rocks in the TIR is due to the high value of the refractive index $n$ (in this case, we calculate in the 8–13 μm range values of $n(\lambda)$ between 2.1 at $\lambda = 8.8$ μm and 8.3 at $\lambda = 9.5$ μm, with an average value of 4.5). The refractive index of water in the 8–13 μm range is ∼1.3—higher than that of air (1.0)—therefore, a thin water layer on top of the rock surface decreases the refractive-index difference (i.e., the electromagnetic wave velocity impedance mismatch) between the rock and the external space. It is crucial that water is weakly absorbing in the 8–13 μm range: for example, at $\lambda \simeq 6$ μm liquid water has a spectral emittance peak that would completely reverse the TIR imaging contrast (higher reflectance for rocks covered by a water layer); however, this behavior cannot be observed with standard TIR cameras because it is outside the atmospheric transparency window of 8–13 μm. Orange lines in Figure 3 identify the atmospheric transparency window, which defines the spectral range of integration $\Delta\lambda$.

Finally, one should notice that, in all cases where emissivity dominates over reflectivity in Equation (4), the effect of water layers is probably not observable, as the spectral emittance of water in the 8–13 μm range is negligible if compared to that of rocks (compare the dotted curve in Figure 3a with the dotted curve in Figure 3b).

## 4. Experimental Results

Thermal images have been acquired with a high-performance TIR camera (Gobi 640 by Xenics) with $640 \times 480$ pixels. The sensor is a microbolometer array made of amorphous silicon sensing elements, with pixel pitch of 17 μm, processed on a complementary metal-oxide semiconductor (CMOS [33]) readout integrated circuit (ROIC [34]) featuring an analog-to-digital converter (ADC) that provides fully digital images in raw intensity format, i.e., ADC counts per pixel. The camera mounts a germanium lens-objective with a focal length of 18 mm, and f/1 optics corresponding to a horizontal and vertical field of view full-angle of 33° and 25°, respectively (see Figure 2d). The spectral range of sensitivity of the microbolometer array is limited by a long-pass filter to TIR $\lambda > 8$ μm, and it is effectively limited to $\lambda < 13$ μm by atmospheric absorption.

The data acquisition time was 11:38 am on 27 October 2021, in clear sky conditions at the terrestrial coordinates of 43.910538°N, 11.807625°E. The ambient temperature value was 14°C and the position of the sun at the time of image acquisition was 30.23° of altitude and 156.84° of azimuth. The approximate azimuthal angle between this solar radiation direction and the scarp plane (which is oriented at 15° from the north–south direction, see Figure 1c) was then 40°. The distance of the camera from the landslide crown varied from∼300m at the southern end to∼540 m at the northern end (higher elevation point, 280 m above the camera). The lateral sizes of the elements of the scarp imaged by each pixel varied between 0.3 and 0.6 m, depending on view angles, distance, and elevation. Due to the high elevation and short distance of the camera from the scarp, many specular reflection paths at the scarp surface existed, connecting the surrounding environment to the camera (Figure 3b). A typical resulting image, acquired with a frame rate of 19 Hz (0.53 s per frame, suitable for real-time monitoring), is shown in Figure 4. In the left panel, we show the image in a commonly used temperature color scale that assigns a temperature value to each pixel using Equation (1). While the TIR image can certainly be utilized, e.g., to qualitatively detect structural features, this temperature scale is not meaningful in the present case because the reflected power $R$ dominates over the emitted power $J$, see Equation (4). We therefore did not further use the temperature scale in this work. In the right panel we therefore show the raw ADC values. As expected, the TIR signal from the sky is null and the vegetation reflects far less than the rocks due to diffuse light scattering that takes place also at TIR wavelengths. Vegetation areas (approximately corresponding to

the greenish pixels in the left panel) were identified by comparison with the visible image in Figure 2c). The interesting portion of the image is the rocky scarp not covered by vegetation (approximately corresponding to the reddish pixels in the left panel). In the rocky scarp, the different sandstone and clay strata are apparent, as their real size is larger than the imaged pixel size of 0.3–0.6 m depending on the view angle. The foot of the scarp close to the lowest white dot is very evident in the TIR images but it is just the dividing line between the rocky scarp and its own debris. Beyond the "fast" TIR intensity modulation due to the strata, which can be rigorously canceled as we shall see in the next section, a much weaker modulation of the TIR signal intensity summed over larger sub-portions of the rocky scarp is observed. The sub-portions are difficult to see with the naked eye, therefore we resorted to multiscale segmentation to highlight it, but we anticipated that, in some cases, they would correspond to overhanging strata with a surface orientation different to that of the rest of the scarp, and in other cases a contrast independent of the orientation of the strata was observed. We note that the average surface orientation with respect to the sunlight direction is uniform across the scarp, resulting in similar solar irradiation levels. The TIR component of the sunlight may then add up to the environmental thermal emission and be reflected towards the TIR camera, but it will not produce a shadow-like contrast among the different scarp sub-portions (except the overhanging strata). At the moment, we cannot state whether the TIR component of the sunlight is dominant over the environmental TIR emission: answering this question will be crucial in establishing whether the proposed landslide monitoring technique can be employed in darkness conditions or not.

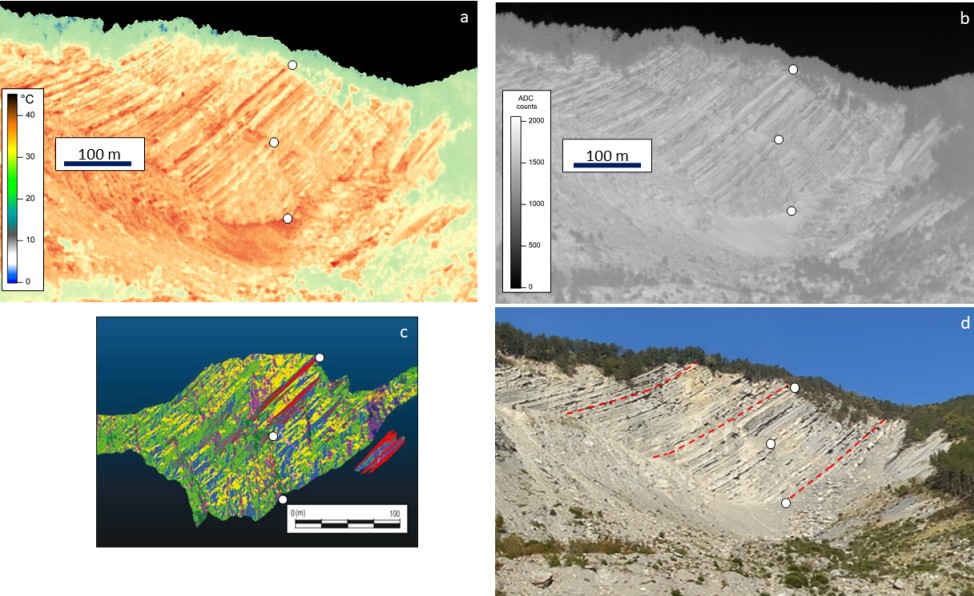

**Figure 4.** (**a**) Thermal image represented in the false-color temperature scale loaded in the camera software. The sky pixels have low values and have been manually set to black. (**b**) Thermal image in grayscale, corresponding to the raw image data in analog-to-digital-converter (ADC) counts. Note the stacking fault in the middle of the scarp, almost perpendicular to the strata, which is very clearly visible in the TIR images (**c**) point-cloud image of the same rocky scarp, obtained by stroboscopic imaging with an unmanned aerial vehicle (reproduced from Ref. [35]): the different colors highlight the discontinuity systems enclosed within rock joint lines, which may be taken to correspond to different homogeneous rocky areas (e.g., the stripes in red color represent the sandstone overhangs mentioned in the paper text). The angle of observation is slightly different in the thermal/visible image and in panel c, therefore we added three reference points to all panels (white dots). (**d**) visible image of the scarp showing the three different members of the marly-arenaceous formation separated by dashed red lines, according to [36].

## 5. Image Analysis

The proposed model is mainly oriented to obtain a new representation of the thermal image that could be analyzed by human experts with the naked eye. Such a representation should then be processed by the human visual system [37] but it must have been built through an objective procedure. As a matter of fact, processing the TIR image resembles the problem of the detection of slightly visible regions in an image, which can be successfully solved by employing some principle of the human visual system, such as contrast sensitivity and multiscale filtering [37]. The former is the ability to distinguish an object (luminance) from its surrounding background; the latter represents one of the basic models for human vision and corresponds to the ability to perceive or disregard details. It turns out that there are three main phases that characterize the proposed approach:

- **Spatial regularization.** A change of scale of the image under study oriented to eliminate a spurious single-pixel classification that confuses the human eye. This step can be further stressed using more than one scale where the resulting matrices are then multiplied by each other (multiscale regularization);
- **False intensities distribution regularization.** A histogram equalization of the resulting image in order to make the image contrast 'uniform' to be better analyzed by the human visual system;
- **Intensity multiple thresholding.** A suitable segmentation of the resulting image is then necessary to partition the resulting image into homogeneous regions characterized by the same physical properties.

In the following, a short presentation of each phase will be given.

**Spatial regularization.** The change of scale of the input image can be seen as a simple application of the Multiresolution Analysis theory. The latter involves the formal definition of how to produce different scales of a given function that are correlated with each other by some mathematical properties [38]. This representation is oriented to highlight specific details of a given signal in agreement with the pioneering studies on multiresolution pyramids by Burt and Adelson [39], and the formal construction of orthogonal wavelets [40].

Coarsely speaking, a given function $f$ at a resolution $2^{-j}$ can be seen as a (discrete) grid of samples where local function averages are considered—the size of the average domain is proportional to $2^j$. It turns out that a multiresolution approximation of $f$ is composed of different and embedded grids. Very often, this operation becomes more intuitive by considering each one of these grids (say that at resolution $2^{-j}$) as the orthogonal projection on the space $V^j \subset L^2(R)$. The latter includes all possible approximations at the resolution $2^{-j}$. Hence, starting from a given function $f$, its approximation $f_j$ at resolution $2^{-j}$ is the projection on the space $V^j$ constrained to minimize the following quantity: $||f - f_j||$. Usually, this operation can be achieved by convolving $f$ with a dilated and translated version of a scaling function $\Phi$. The Haar scaling function [38] and just a few smoothed versions of $f$ were used in this study. The Haar scaling function computes local averages of $f$ with on a fixed window, different for each multiresolution level. Keeping in mind that in our study case $f \in \mathbf{R}^2$, as it is a 'thermal image' $I(x, y)$, we can consider $I(x, y)$ at a given resolution $2^{-j}$, i.e., $I_j(x, y)$. The desired resolution is the one at which local details (e.g., the arenaceous strata) are disregarded while contrast sensitivity for the wet region is enhanced, as shown in Figure 5a.

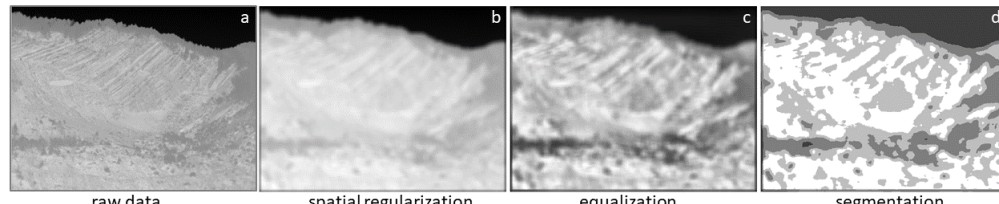

**Figure 5.** (**a**) Raw TIR image data; (**b**) low-resolution version of panel a; (**c**) output of the histogram equalization—regions having homogeneous local luminance values are now emphasized; (**d**) segmented image using a choice of four bins.

**False intensities distribution regularization** and **Intensity multiple thresholding**. On such a function, two additional operations were performed: histogram equalization and a suitable segmentation. The former aims at emphasizing homogeneous regions, the latter aims at separating those regions. Their combined use resembles a non linear segmentation. Histogram equalization of an $L$ levels (i.e., $\log_2(L)$ bits) image $I(x, y)$ consists of defining a new image $J$ having a flat histogram [41], i.e.,

$$J = T(I),\tag{6}$$

where $T$ is a proper transformation that depends on the discrete ($L$-levels) cumulative distribution function of $I$. A simple rescaling operation maps values back to the original range of values:

$$\bar{J} = J \cdot (\max\{I\} - \min\{I\}) + \min\{I\} = J \cdot (L - 1).\tag{7}$$

Figure 5b shows the equalized low-resolution image.

The last step is based on scalar segmentation [41], whose objective is to reduce the set of possible image values, as shown in Figure 5d. A segmentation function, or quantizer $Q(x)$, is:

$$Q : \mathbf{R} \to C,\tag{8}$$

where $\mathbf{R}$ is the real axis while $C \equiv \{y_1, y_2, ..., y_N\} \in \mathbf{R}$ is the codebook, with $y_1 < y_2 < ... < y_N$ and $\Delta_k \equiv (y_{k-1}, y_k]$ the segmentation bins. $C$ has finite cardinality, i.e., $|C| = N$. In the present study, a uniform and scalar quantizer was considered. Hence, the range for image intensity value $[a, b]$ was split into $K$ sub-intervals $\{(y_{k-1}, y_k]\}_{1 \leq k \leq K}$ having the same size and

$$\forall x \in (y_{k-1}, y_k], \qquad Q(x) = x_k,\tag{9}$$

with $x_k \in (y_{k-1}, y_k]$—$x_k$ is usually set equal to the middle value of $(y_{k-1}, y_k]$. The choice of how many scale levels as well as how many segmentation bins to select must be adapted to each study case. Figure 5 represents the intermediate steps of the image segmentation analysis (regularization, equalization, quantization) from the raw data of the thermal image. These steps "simulate" image processing steps that are believed to happen in the human brain (not in the eye itself) to transform an image into useful information for decision-making. For the final image of the present work (Figure 6), we employed the finer multiscale segmentation result shown in Figure 7. Image analysis can be refined in the future by fine tuning the number of scales to be used for image regularization, and the number of segmentation bins. Methods based on the minimum description length are particularly well suited for this purpose.

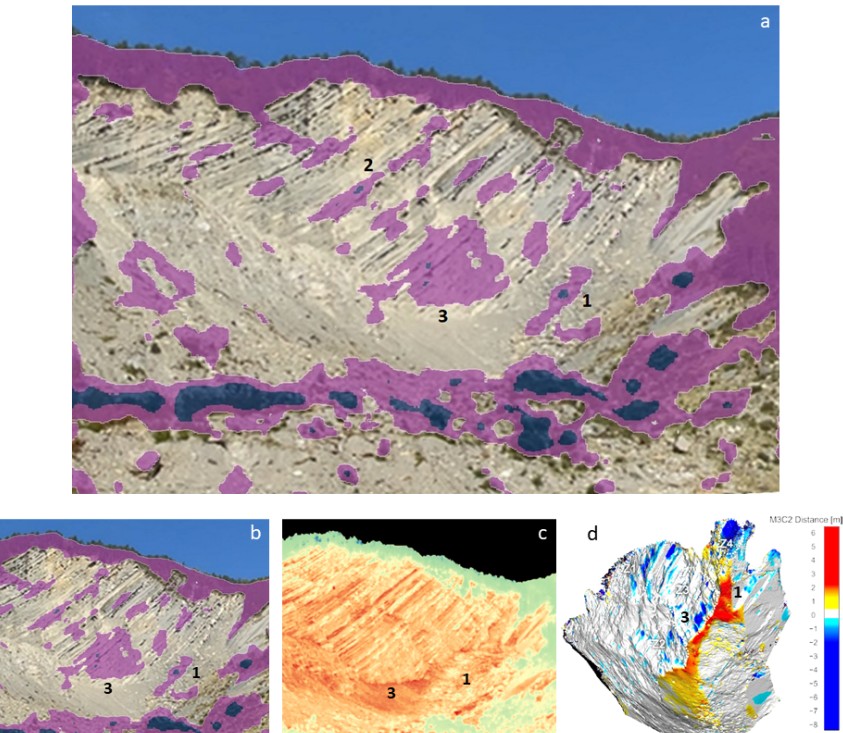

**Figure 6.** Visible image of the upper scarp with overlapping of multiscale segmentation map in false purple color scale: only the two lowest levels are shown, while the two highest levels are set as a transparent layers. The result is an overlayed map of areas of the rocky scarp with an anomalously low TIR signal, relevant for rockfall hazard monitoring. The lowest level in dark purple almost entirely corresponds to vegetation-covered areas, for which our physical model does not hold, and therefore they should not be considered as relevant areas but rather simply as low TIR reflectance areas. (**b**) Zoom in of a detail in the image (**a**), (**c**) the same FOV of image b in false color in TIR range, (**d**) a stroboscopic point-cloud performed by UAV photogrammetry, image from [36].

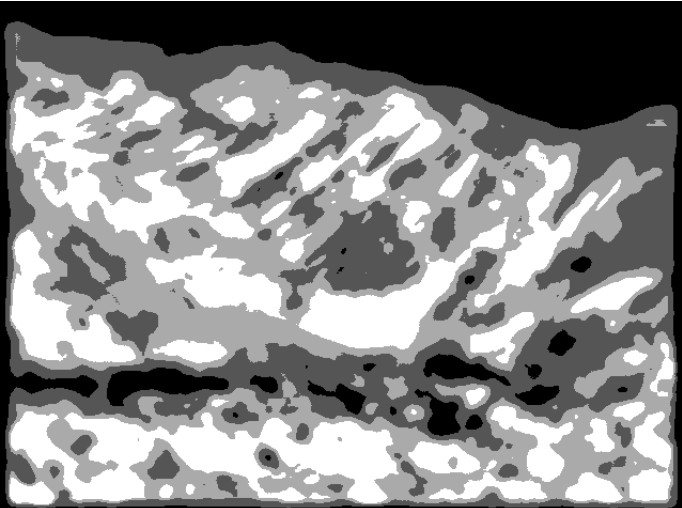

**Figure 7.** Output of the proposed image analysis procedure where four scales have been properly combined in the spatial regularization step, before applying histogram equalization. A more detailed segmentation is provided in this case.

## 6. Interpretation of Results

The image in Figure 6 can be considered the final result of our study. The two lowest levels of the chosen multiscale segmentation are identified as vegetation areas (to be

discarded) and monitoring target sub-portions of the scarp that produce an anomalously low TIR signal. The two highest levels are interpreted as normal scarp sub-portions not relevant for hazard monitoring purposes, and therefore they have been set as transparent layers in Figure 6. A baseline was defined within the output of the segmentation algorithm, which included all pixels with uniform intensity value and spatial distribution so sparse that they could not be identified as a single geological unit, and therefore as a specific hazard cause. For this reason, the two highest segmentation levels were made transparent because their TIR image provided no information, at least within our segmentation analysis.

The lowest segmentation level (dark blue zones in Figure 6a,b) identifies vegetation areas at the foot of the rocky scarp. Vegetation is assumed to reflect far less TIR intensity in the specular direction than the rocky scarp due to scattering from the "rough" surface constituted by leaves, branches etc. A few very small dark blue zones also appear on the rocky scarp, at the center of large purple areas, a natural effect of multiscale segmentation which could be eliminated by an appropriate contour logic if necessary.

The second-lowest segmentation level (purple zones in Figure 6a,b) identifies sub-portions of the scarp that are relevant for hazard monitoring because their TIR signal is anomalously low. The first class of relevant areas are those with high humidity. According to our physical model, thin layers of liquid water (hundreds of μm thickness, compatible with local humidity of the rock surface) can decrease the TIR reflectance substantially. The valley marked with the number "1" in Figure 6 and the rocky areas under the vegetation are very likely to be more humid in the morning hours, indeed. The connection between humidity and rockfall hazard is yet to be established in full in this case, but it is likely considering the landslide history.

A second class of relevant areas obviously corresponds to the overhanging arenaceous strata, one of them marked with number "2" in Figure 6a). Following our physical model, reflectance is expected to dominate over emittance, unless the surface orientation angle of some sub-portions is different from the average scarp orientation. The orientation angle of these overhanging strata is certainly very different from the average scarp orientation. We expect therefore that the TIR power reflected by these sub-portions of the scarp would not reach the TIR camera but rather be reflected outside the field of view. This TIR contrast mechanism can be considered a useful marker of overhanging strata that are known to be the origin of rockfall events.

The wide purple zone in the middle of the scarp marked with "3" in Figure 6 cannot be trivially identified as humid or overhanging. Looking at the detailed geological studies of the upper scarp [26,27], we have identified this area as a freshly cleaved site of a large rockfall event. It is possible that freshly cleaved minerals present a different TIR spectral signature if compared to the weathered minerals that we considered up to now (including the argillaceous sandstone sample from the ECOSTRESS library). The evidence for changes in the TIR reflectance spectra of minerals with weathering can be found in the literature ([42–44]). In the specific case of the argillaceous sandstone of the Poggio Baldi landslide, high-resolution IR spectroscopy laboratory analysis of fresh and weathered samples is planned and will be published in the future.The wavelength-integrated spectral reflectance of Equation 5 could be either higher or lower for freshly cleaved minerals. This hypothesis requires further studies, including laboratory TIR spectroscopy with Fourier Transform infrared (FTIR) spectrometers of local rock samples.

The structural characterization of the freshly cleaved surface section labelled with the number "3" in Figure 6a–c is provided in Figure 6d from [36] in the form of a point-cloud image difference between two acquisitions at distant times (3 years) with the algorithm Model to Model Cloud Comparison (M3C2) of Ref. [45]. This image highlights the rock volumes lost and gained by different parts of the slope; the scarp sub-portion labelled with "3" in Figure 6 has been identified as a freshly cleaved surface because of high volume loss. A similar identification has been made from terrestrial-laser-scanner (TLS) data in Ref. [27].

Finally, the purple zones in Figure 6a,b corresponding to the vegetation, both above the landslide crown and at the scarp foot, must be discarded from anomaly detection

because they do not correspond to any portion of the rocky wall, nor are they at a different temperature to the rocky wall: they simply reflect less TIR radiation because of scattering by branches, leaves, and trunks.

## 7. Discussion and Conclusions

In this work, we present a first study of an automatic remote-sensing method for monitoring landscapes. At present, our thermal imaging-based sensing method could be used for the preliminary screening and extensive evaluation of rocky scarps, with the aim of producing a potential hazard map. Based on this map, the hazard map can be complemented with local observation with different remote sensing techniques (e.g., laser scanner, unmanned aerial vehicle analysis etc.) and, if needed, with invasive techniques (e.g., stress sensors). To reach the goal of the preliminary screening of rocky scarps using thermal imaging, it is essential to identify, emphasize and extract the informative features, to detect the possible artifacts, and the information-lacking background pixels. Our image segmentation algorithm has been designed to perform the above tasks. Finally, the automated algorithm has internal parameters that must be properly tuned to replicate as much as possible the naked-eye analysis of the thermal images typically made by a human expert.

In the future, with the acquisition of time series in conjunction with actual rockfall observation over time, it should be possible to precisely define alert threshold values that may eventually enable early warning signals based on day-and-night thermal imaging in a completely automated way. It should be noted that vegetation-covered areas cannot be analysed with the specular reflectance model employed in this work, as their infrared response is dominated by diffuse scattering. It is also difficult to envisage the application of our method to the investigation of slow-moving debris masses without a relatively flat, well-oriented surface.

We have applied an advanced image analysis technique, based on multiscale segmentation, to enhance the contrast of thermal infrared (TIR) images of the upper scarp of a landslide taken at a distance of 500 m. We have found that the TIR image contrast analysis can highlight different physical mechanisms behind TIR contrast: (i) a different local orientation of the rocky wall if compared to the average scarp surface orientation, due to the fact that reflectance will in general dominate over emittance of rocky walls, but more so for properly oriented scarp sub-portions; (ii) a different grade of humidity of scarp sub-portions, because, according to our physical model of the optical constants in the TIR wavelength range, even a surface water layer of 100 µm thickness can decrease the TIR signal to an extent that can be observed by a high performance TIR camera; (iii) a different mineral content in scarp sub-portions, due to the high sensitivity of TIR spectral reflectance to the specific mineral oxidation state, which is related to the exposure time to weathering agents. All these TIR imaging contrast mechanisms constitute promising conceptual tools for future automated landslide monitoring based on TIR imaging, where the presented image analysis algorithms would be fully automated and their selected threshold would be connected to alarm signal generators, also fed by different complementary remote sensing technologies.

**Author Contributions:** research design: P.M. and M.O.; image acquisiton: A.M.; data analysis: A.M., M.O., D.V., V.B. and P.M.; paper writing: M.O., V.B. and A.M.. All authors have read and agreed to the published version of the manuscript.

**Funding:** The authors acknowledge funding from Sapienza University of Rome - Ricerca d'Ateneo 2020 - grant no. PH120172B3E3E36D.

**Data Availability Statement:** All data are available on reasonable request.

**Conflicts of Interest:** The authors declare no conflict of interest

.

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
