# Peer review of "Enhancing the Thermal Images of the Upper Scarp of the Poggio Baldi Landslide (Italy) by Physical Modeling and Image Analysis"

_remotesensing, doi:10.3390/rs15040907_

Round 1

Reviewer 1 Report

The authors present new methods for physical interpretation and mathematical treatment of the imaging contrast observed in thermal images of the rocky upper scarp of the Poggio Baldi landslide. This work is of good structure and writing, and I have several comments below:

1.     In the Introduction section, the author is suggested adding the field-scale or lab-scale application of TIR imaging in geological hazard monitoring and pointing out the potential problems that still exist. Then, tell the reader how this paper tried to solve those problems. 

2.     What about the feasibility of the proposed methods if a steep scarp is made not only of clean rocks but also of a high density of vegetation? How to exclude environmental thermal sources apart from the targeted goals?

3.     The thermal characteristics of a rocky upper scarp of a landslide can be noticeable. However, what about the feasibility of the proposed methods in a slow-moving landslide? In general, the thermal characteristics of this type of landslide are less obvious, especially deep-seated landslides. The author is suggested to discuss this point.  

4.     Lines 362 to 363, “It is possible that freshly cleaved minerals present a different TIR spectral signature if compared to the weathered minerals that we considered up to now (including the argillaceous sandstone sample of the ECOSTRESS library).” It is interesting, but is there evidence for this point? What’s the meaning of this finding?  

5.     How about the reliability of the TIR image combined with the proposed methods? In geological engineering, this technology will be influenced by temperature inhomogeneity, surface topography, lousy weather, stratigraphic properties, etc.

Author Response

Thank you for your comments. Please find attached our Reply point-by-point, your text in balck color, our replies in blue color.

Author Response

(The authors gave the same response as above.)

Reviewer 3 Report

The manuscript entitled “Enhancing the thermal images of the upper scarp of the Poggio Baldi landslide (Italy) by physical modeling and image analysis” deals with the characterization of a rock mass by infrared thermography, attempting to enhance the contrast of thermal infrared images by an advanced image analysis technique, based on multiscale segmentation.

The topic is interesting from the scientific point of view according to the increasing utility that infrared thermography is bringing in the study of rock masses. Nevertheless, I personally found this study incomplete, because no data validation is provided, and the interpretation of thermal images seems superficial to me. Moreover, some potentially relevant rock mass features suggested by Figure 4 (see comments below) have not been highlighted by this study.

My main comments are listed below:

1.     Introduction: a literature overview on the recent technologies is reported, but there is a lack of literature experiences on the recent technological advance on use of TIR for landslide mapping and monitoring. A brief state of the art should be presented.

2.     Section 2: besides the bedding attitude, no further rock mass datum is presented. This is a weak point of such an experimental study, because remotely sensed data should be compared to reliable field information for validation.

3.     Figure 4: the figure quality is not high enough. Scale bars seems to be randomly placed, even in the middle of a figure (right panel). It would be also useful to provide a photograph in the visible range of the rock mass portion framed (like the one in Figure 2).

4.     Figure 4: the false-color temperature scale image shows some relevant features, which have not been addressed by the authors and deserve to be verified in field and commented. From what I can observe in the small figure frame, there are at least 2 or 3 potential structural features, which seem to have interesting implication on both the slope morphology and geomechanics. Moreover, what will be later on the text defined as a “freshly cleaved site of a large rockfall event” is clearly recognizable in this image and would have deserved specific comments and interpretations (to be then verified in the field), even with reference to some thermal features that the image suggests and that be correlated to structural elements.

5.     Section 4: No ambient temperature value and rock slope exposure, with respect to the solar radiation, is reported herein.

6.     Section 4: authors state that “this temperature scale is not meaningful in the present case…”. According to my previous comment 4, I think that some relevant data could be derived from that image.

7.     Fig. 5: I failed to catch the utility of this figure.

8.     Section 6: to highlight the utility of their approach, authors should have provided a comparison between a TIR color-scaled image, and the image presented in Figure 6. Their discussion should be focused on what the multiscale segmentation approach can highlight compared to the typical TIR image. I believe that Figure 6 alone, along with its description provided in section 6, are not enough to be defined the results of the study. In fact:

- vegetation and humidity zones can be well mapped by TIR according to a false-color temperature scale image;

- overhanging rock mass parts can be highlighted by the analysis of a false-color temperature scale TIR image;

- the “freshly cleaved site of a large rockfall” was already visible and well definable by the analysis of figure 4.

9.     Section 6 looks more a paragraph on the interpretation of results rather than a discussion. A discussion should provide critical comments on achieved results, to support interpretation, even involving the knowledge already available in the literature.

According to these major considerations, I think this manuscript is not acceptable for publication in Remote Sensing journal.

Best regards

Author Response

(The authors gave the same response as above.)

Round 2

Reviewer 3 Report

The manuscript entitled “Enhancing the thermal images of the upper scarp of the Poggio Baldi landslide (Italy) by physical modeling and image analysis” has been revised by the Authors, who took into account my comments clarifying some aspects. In particular, authors have better explained their aims and point of view, adding some key datum to the manuscript, even by introducing a recent published work on the same study area.

I believe that a further effort should be done to achieve a scientific work useful for the international community working in this field. Please, find below my suggestions:

1.     Introduction: although few references were added by the Authors, I think that the most recent developments on the use of infrared thermography to rock masses were not acknowledged. By performing a quick research on Remote Sensing  journal (thermography+rock) the following additional papers should be mentioned in my opinion:

- Mineo et al., 2022. https://doi.org/10.3390/rs14030473

- Loche et al., 2021. https://doi.org/10.3390/rs13071265

- Frodella et al., 2020. https://doi.org/10.3390/rs12050892

2.     Introduction, lines 91-99: this text sounds more like a discussion. I suggest to move it in a discussion paragraph to be prepared according also to my following comments.

3.     In figure 4a, there are some linear anomalies possibly related to 45° planes crossing the rock mass (1 of these crosses the intermediate dot and corresponds to one of the planes forming the wedge on the top of rock mass). Some throws between bedding planes are also suggested, but it might be only a matter of optical illusion due to perspective issues. Authors are kindly invited to clarify is this feature.

4.     In figure 4a the foot of the area defined “freshly cleaved surface” seems to be undercut. Is this confirmed? If so, this would represent a key instability feature that should be mentioned.

5.     The image provided in the point-by-point reply, taken from ref. 42, should be inserted in the text because it provides some clarifications for a reader who has never been in that study are. Also, some images taken from ref.42, such as figures 2 and 7, should be provided in this manuscript as they bring some clarifications from the geostructural point of view.

6.     Lines 381-382: what stated in brackets should be supported by a literature reference. There is a recent work published by Vivaldi et al., 2022 (DOI 10.1007/s10346-022-01970-z), where vegetation is reported, in some cases, with a higher surface temperature than the slope.

7.     I agree with renaming section 6, but I think that a discussion paragraph should be provided. In particular, potential and limitation of the presented approach should be reported. Moreover, I think that Authors should provide a figure on the comparison between raw and segmented TIR images supporting their discussion (the same figure that they reported in the point-by-point reply). The discussion section could also contain the considerations that Authors reported as a reply to the reviewer talking about this comparison

I think that the manuscript has been improved after the first revision stage. Nevertheless, some more points should be addressed before considering it suitable for publication.

Best regards

Author Response

See attached pdf file. We have added figures and references to the manuscript as requested.

Round 3

Reviewer 3 Report

Authors have revised their manuscript by taking into account the suggestions and an improved version has been achieved. 

Please note that the Discussion section should be numbered and placed before the Conclusion section. In this regard, authors could also think of merging the discussion and conclusions into a single paragraph, likely titled "Discussion and Conclusions", where Authors should make the last effort to highlight the importance of the study, stressing both the novelty and the limitations, even in the perspective of future studies.

Best Regards  

Author Response

We have merged the "discussion and conclusion" sections in a single paragraph. We have reworded the paragraph, highlighting the novelty, the potential, the limitations and the outlook of our novel infrared image analysis technique.